# Back Transcription as a Method for Evaluating Robustness of Natural Language Understanding Models to Speech Recognition Errors

**Marek Kubis**
**Paweł Skórzewski**
Adam Mickiewicz University, Poznań
ul. Uniwersytetu Poznańskiego 4
61-614 Poznań, Poland
{mkubis,pms}@amu.edu.pl

**Marcin Sowański**
**Tomasz Ziętkiewicz**[*]
Samsung Research Poland
Plac Europejski 1
00-844 Warsaw, Poland
{m.sowanski,t.zietkiewic}@samsung.com

## Abstract

In a spoken dialogue system, an NLU model is preceded by a speech recognition system that can deteriorate the performance of natural language understanding. This paper proposes a method for investigating the impact of speech recognition errors on the performance of natural language understanding models. The proposed method combines the back transcription procedure with a fine-grained technique for categorizing the errors that affect the performance of NLU models. The method relies on the usage of synthesized speech for NLU evaluation. We show that the use of synthesized speech in place of audio recording does not change the outcomes of the presented technique in a significant way.

## 1 Introduction

Regardless of the near-human accuracy of automatic speech recognition in general-purpose transcription tasks, speech recognition errors can still significantly deteriorate the performance of a natural language understanding model that follows the speech-to-text module in a conversational system. The problem is even more apparent when an automatic speech recognition system from an external vendor is used as a component of a virtual assistant without any further adaptation. The goal of this paper is to present a method for investigating the impact of speech recognition errors on the performance of natural language understanding models in a systematic way.

The method that we propose relies on the use of back transcription, a procedure that combines a text-to-speech model with an automatic speech recognition system to prepare a dataset contaminated with speech recognition errors. The augmented dataset is used to evaluate natural language understanding models and the outcomes of the evaluation serve as a basis for defining the criteria of

NLU model robustness. Contrary to conventional adversarial attacks, which aim at determining the samples that deteriorate the model performance under study (Morris et al., 2020), our method also takes into consideration samples that change the NLU outcome in other ways. The robustness criteria that we formulate are then used to construct a model for detecting speech recognition errors that impact the NLU model in the most significant way.

The proposed method depends on speech processing models, but it does not rely on the availability of spoken corpora. Therefore, it is suitable for inspecting NLU models for which only textual evaluation data are present. It makes use of the semantic representation of the user utterance, but it does not require any additional annotation of data. Thus, the dataset used for training and testing the NLU model can be repurposed for robustness assessment at no additional costs. For illustration, we decided to apply the presented method to Transformer-based models since they demonstrate state-of-the-art performance in the natural language understanding task, but the method does not depend on the architecture of the underlying NLU model. The limitations of our approach are discussed at the end of the paper.

## 2 Related Work

Data augmentation is a commonly employed method for improving the performance of neural models of vision, speech and language. Ma (2019) developed *nlpaug*, a tool that encompasses a wide range of augmentation techniques for text and audio signal. Morris et al. (2020) presented a framework for adversarial attacks for the NLP domain, called TextAttack, that can be utilized for data augmentation and adversarial training. The framework has a modular design – attacks can be built from four components: a goal function, a set of constraints, a transformation, and a search method. It also provides out-of-the-box implementations for

---

[*]The author performed the work while being affiliated with both organizations.

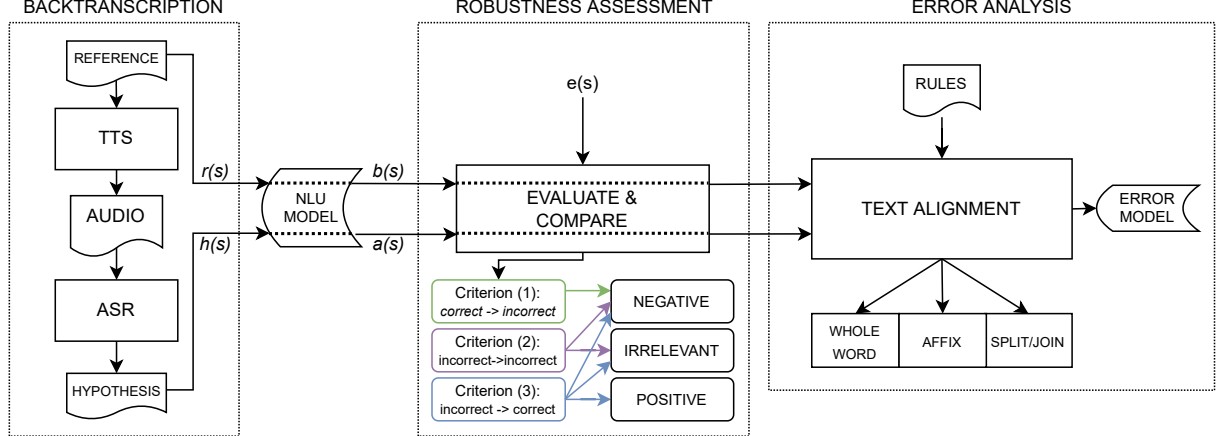

Figure 1: The proposed method that consists of back transcription, robustness assessment, and error detection.

several popular attacks described in the literature.

Back translation (Sennrich et al., 2016) is a technique developed to improve the performance of neural machine translation. In this method, additional monolingual data for training the model are obtained by translating utterances from the target language to the source language.

The first experiments with augmenting ASR data with text-to-speech tools were conducted by Tjandra et al. (2017). They generated speech data from unpaired texts to create a paired speech-text corpus. Hayashi et al. (2018) proposed a novel data augmentation method for end-to-end ASR. Their method uses a large amount of text not paired with speech signals in a back-translation-like manner. The authors claim to achieve faster attention learning and reduce computational costs by using hidden states as a target instead of acoustic features. Laptev et al. (2020) built a TTS system on an ASR training database, then synthesized speech to extend the training data. The authors distinguish two main approaches to ASR: hybrid (DNN+HMM) and end-to-end. They noted that both approaches achieve similar performance when there is a large amount of training data (Lüscher et al., 2019), but hybrid models perform better than end-to-end models when the amount of data is smaller (Andrusenko et al., 2020). Park et al. (2021) construct a parallel corpus of texts and their back-transcribed counterparts for the purpose of training a post-processor for an ASR system. They show that their method is effective in correcting spacing, punctuation and foreign word conversions.

Recently, there have been several papers addressing the issue of the robustness of natural language understanding systems to various types of input errors. Liu et al. (2021) defined three aspects of NLU robustness (language variety, speech characteristics, and noise perturbation) and proposed a method of simulating natural language perturbations for assessing the robustness in task-oriented dialog systems. The influence of the noisy text input on the NLU performance was also studied by Sengupta et al. (2021), who tested intent classification and slot labeling models' robustness to seven types of input data noise (abbreviations, casing, misspellings, morphological variants, paraphrases, punctuation, and synonyms). Peng et al. (2021) created a benchmark for evaluating the performance of task-oriented dialog models, designed to favor models with a strong generalization ability, i.e., robust to language variations, speech errors, unseen entities, and out-of-domain utterances.

## 3 Method

The proposed method of evaluation consists of three stages: the execution of the back transcription procedure that transfers NLU data between text and audio domains, the automatic assessment of the outcome from the NLU model on a per-sample basis, and the fine-grained method of inspecting the results with the use of edit operations (see Figure 1). The method relies on the availability of an NLU dataset containing user utterances along with the semantic representation of the uttered commands. We do not assume any particular annotation scheme for the NLU commands. However, in the experiments discussed in Section 4, we work with the dataset that provides conventional annotations for domains, intents, and slots.

## 3.1 Back Transcription

The back transcription procedure applied with respect to the NLU dataset consists of three steps. First, textual data are synthesized with the use of a text-to-speech model. Next, the automatic speech recognition system converts the audio signal back to text. In the last step, both the input utterances and the recognized texts are passed to the NLU model to obtain their semantic representations.

The NLU commands are tracked in consecutive steps. As a result, we obtain an augmented NLU dataset providing the following data for each sample $s$:

1. $r(s)$: the reference text that comes from the initial NLU dataset;

2. $h(s)$: the hypothesis, i.e. the $r(s)$ text synthesized with the text-to-speech model and transcribed with the automatic speech recognition system;

3. $e(s)$: the expected outcome of the NLU model for $r(s)$ as given in the initial NLU dataset;

4. $b(s)$: the outcome of the NLU model for $r(s)$ (i.e. the NLU result for the text *before* back transcription);

5. $a(s)$: the outcome of the NLU model for $h(s)$ (i.e. the NLU result for the text *after* back transcription).

## 3.2 Robustness Criteria

A simple method for coarse-grain assessment of NLU robustness relies on measuring performance drop with respect to the commonly used metrics such as accuracy for intent classification or F-score for slot values extraction. This is a widely accepted practice in the case of adversarial attacks (Morris et al., 2020); however, such a procedure does not distinguish specific cases that arise in the comparison of the model outcomes. Let us divide samples that differ in NLU outcomes obtained for reference utterances and their back-transcribed counterparts along three criteria:

1. $C \rightarrow I$

    A correct result obtained for the reference text is changed to an incorrect one in the case of the back-transcribed text, i.e. $b(s) = e(s) \wedge a(s) \neq e(s)$.

2. $I \rightarrow I$

    An incorrect result returned for the reference text is replaced by another incorrect result in the case of the back-transcribed text, i.e. $b(s) \neq e(s) \wedge a(s) \neq e(s) \wedge b(s) \neq a(s)$.

3. $I \rightarrow C$

    An incorrect result obtained for the reference text is changed to a correct result in the case of the back-transcribed text, i.e. $b(s) \neq e(s) \wedge a(s) = e(s)$.

The first category is always considered to have a negative impact on the robustness of the NLU model. However, with respect to $I \rightarrow I$ and $I \rightarrow C$ categories of samples, alternative options can be considered. For $I \rightarrow I$ samples, it is reasonable to treat them as negative if we want to obtain the definition of robustness that penalizes changes. It is also sensible to consider them to be irrelevant since such samples do not affect the performance of the NLU model. $I \rightarrow C$ samples, once again, can be considered to be negative if we want to penalize all changes. They can be treated as irrelevant, making the definition of robustness unaffected by the changes that improve the performance of the NLU model. Finally, they can also be considered to have a positive impact on the robustness of the model since they improve the NLU performance.

## 3.3 Problems of $\Delta$-Measurements

A common practice of measuring the difference in accuracy before and after back transcription treats $I \rightarrow C$ samples as positive and ignores $I \rightarrow I$ samples. Such a procedure underestimates the impact of $C \rightarrow I$ samples on the NLU module due to the performance gain introduced by $I \rightarrow C$ samples. It also does not track $I \rightarrow I$ changes which can deteriorate the behavior of downstream modules of a dialogue system that consume the outcome of the NLU model. As we show in Section 4, $I \rightarrow I$ and $I \rightarrow C$ cases account respectively for up to $30\%$ and $10\%$ of all the changes introduced by back transcription. Thus, the decision to ignore or promote them should be a result of careful planning.

The relationship between the outlined categories of changes and the building blocks of the F-score is even more complicated. Let $C_\alpha \rightarrow I_\beta$ denote the change from correct label $\alpha$ to incorrect label $\beta$, $I_\alpha \rightarrow I_\beta$ the change from incorrect label $\alpha$ to incorrect label $\beta$, and $I_\alpha \rightarrow C_\beta$ the change from incorrect label $\alpha$ to correct label $\beta$. Let $TP_l$, $FP_l$, $FN_l$,

| Category | $TP_\alpha$ | $FP_\alpha$ | $FN_\alpha$ | $TP_\beta$ | $FP_\beta$ | $FN_\beta$ | $P_\alpha$ | $R_\alpha$ | $P_\beta$ | $R_\beta$ |
|---|---|---|---|---|---|---|---|---|---|---|
| $C_\alpha \to I_\beta$ | ↓ | = | ↑ | = | ↑ | = | ↓ | ↓ | ↓ | = |
| $I_\alpha \to I_\beta$ | = | ↓ | = | = | ↑ | = | ↑ | = | ↓ | = |
| $I_\alpha \to C_\beta$ | = | ↓ | = | ↑ | = | ↓ | ↑ | = | ↑ | ↑ |

Table 1: Relationship of the building blocks of F-measure and the robustness criteria.

| Name | $I \to I$ | $I \to C$ | Domain ($D$) | Definition |
|---|---|---|---|---|
| $R_{123}$ | negative | negative | $\{s : h(s) \neq r(s)\}$ | $\frac{|\{s : s \in D \wedge b(s) = a(s)\}|}{|D|}$ |
| $R_{13}$ | irrelevant | negative | $\{s : h(s) \neq r(s) \wedge (b(s) = e(s) \vee a(s) = e(s))\}$ | $\frac{|\{s : s \in D \wedge b(s) = a(s)\}|}{|D|}$ |
| $R_{12}$ | negative | irrelevant | $\{s : h(s) \neq r(s) \wedge \neg(b(s) \neq e(s) \wedge a(s) = e(s))\}$ | $\frac{|\{s : s \in D \wedge b(s) = a(s)\}|}{|D|}$ |
| $R_1$ | irrelevant | irrelevant | $\{s : h(s) \neq r(s) \wedge b(s) = e(s)\}$ | $\frac{|\{s : s \in D \wedge b(s) = a(s)\}|}{|D|}$ |
| $R_{123+}$ | negative | positive | $\{s : h(s) \neq r(s)\}$ | $\frac{|\{s : s \in D \wedge (b(s) = a(s) \vee a(s) = e(s))\}|}{|D|}$ |
| $R_{13+}$ | irrelevant | positive | $\{s : h(s) \neq r(s) \wedge (b(s) = e(s) \vee a(s) = e(s))\}$ | $\frac{|\{s : s \in D \wedge (b(s) = a(s) \vee a(s) = e(s))\}|}{|D|}$ |

Table 2: NLU robustness measures.

$P_l$ and $R_l$ denote true positives, false positives, false negatives, precision and recall with regard to label $l$. Table 1 shows how the building blocks of F-measure change due to the changes in NLU outcome between the reference utterances and their back-transcribed counterparts. One can observe that for each category of changes, some components of the F-measure increase (↑) while others decrease (↓) or remain unchanged (=). Therefore, measuring the difference in F-scores between reference utterances and back-transcribed texts leads to results that are difficult to interpret meaningfully.

### 3.4 Robustness Assessment

Proper combinations of the aforementioned categories of NLU outcome changes lead to six alternative robustness measures with their own rationale. We present them in Table 2 and report their values for the NLU models under study in Table 3. The $R_{13+}$ measure, which neglects $I \to I$ changes and treats $I \to C$ changes as positive, is a counterpart of measuring the difference in accuracy for back-transcribed utterances and reference texts. This approach is sufficient for testing an NLU model in isolation, but it does not take into account that the behavior of downstream modules of a dialogue system that consume the outcome of an NLU model can deteriorate due to the change in labeling of incorrect results. Such cases are revealed by $R_{123}$ and $R_{123+}$. The second one promotes changing incorrect outcomes to correct ones, which is rea-

sonable if we assume that the downstream module behaves correctly when presented with a correct input. However, if the downstream module relies on the outcome of NLU regardless of its status [1], then $R_{123}$ should be preferred. $R_{12}$, which penalizes changes between incorrect labels but neglects the impact of $I \to C$ changes, is a rational choice for assessment of an NLU model that precedes a downstream module dedicated to correcting incorrect NLU outcomes such as a rule-based post-processor. $R_1$, which penalizes the drop in accuracy but neglects any other changes, is a metric that tracks the volume of samples that become incorrect due to the use of an ASR system. Therefore, it is suitable for monitoring the regressions of the ASR-NLU pair across consecutive revisions of the ASR model. The penalization of positive changes by $R_{13}$ makes this metric also a reasonable choice for tracking the robustness of NLU models that should act consistently in the presence of reference texts and their transcribed counterparts. This is the case of NLU models that are designed to handle both the input typed by the user and the input that comes from an ASR system. The same holds for $R_{123}$, which, contrary to $R_{13}$, also takes into account the impact of $I \to I$ changes on downstream modules of a dialogue system.

---

[1] A common case in the industrial setting where NLU results are post-processed.

| NLU model | TTS model | $R_{123}$ | $R_{13}$ | $R_{12}$ | $R_1$ | $R_{123+}$ | $R_{13+}$ |
|---|---|---|---|---|---|---|---|
| domain | FastSpeech | 0.8583 | 0.8631 | 0.8704 | 0.8759 | 0.8722 | 0.8778 |
| intent | FastSpeech | 0.8017 | 0.8156 | 0.8131 | 0.8283 | 0.8157 | 0.8309 |
| slots | FastSpeech | 0.3391 | 0.3902 | 0.3470 | 0.4021 | 0.3617 | 0.4199 |
| domain | Tacotron | 0.8903 | 0.9001 | 0.8977 | 0.9080 | 0.8985 | 0.9089 |
| intent | Tacotron | 0.8449 | 0.8616 | 0.8569 | 0.8752 | 0.8589 | 0.8772 |
| slots | Tacotron | 0.3663 | 0.4260 | 0.3744 | 0.4382 | 0.3878 | 0.4539 |

Table 3: NLU models robustness.

| type | name | description | example |
|---|---|---|---|
| whole word operations | del | delete a token | "a" → "" |
| | replace_{r} | replace token with string $r$ | "cat" → "hat" |
| | insert_before_{w} | insert word $w$ before current token | "cat" → "a cat" |
| | insert_after_{w} | insert word $w$ after current token | "cat" → "cat that" |
| affix operations | add_prefix_{p} | prepend prefix $p$ to the token | "owl" → "howl" |
| | add_suffix_{s} | append suffix $s$ to the token | "he" → "hey" |
| | del_suffix_{n} | remove $n$ characters from the end | "cats" → "cat" |
| | del_prefix_{n} | remove $n$ characters from the start | "howl" → "owl" |
| | replace_suffix_{s} | replace last $\text{len}(s)$ characters with $s$ | "houl" → "hour" |
| | sreplace_{s}_{r} | replace substring $s$ with string $r$ | "may" → "my" |
| split/join operations | join_{s} | join tokens using character $s$ | "run in" → "run-in" |
| | split_aftert_{n} | split word after $n$-th character | "today" → "to day" |
| | split_on_first_{c} | split word on first character $c$ | "run-in" → "run in" |
| | split_on_last_{c} | split word on last character $c$ | "forenoon" → "for noon" |

Table 4: Examples of edit operations.

### 3.5 Speech Recognition Errors Detection

To detect speech recognition errors that deteriorate the robustness of the NLU model in the most significant way, we determine the differences between the reference texts and their back-transcribed counterparts and confront them with the impact caused by the change in the NLU outcome. For identifying the differences between reference and back-transcribed utterances, we align them with the use of the Ratcliff-Obershelp algorithm (John W. Ratcliff, 1988). Alignment identifies spans of both texts which are different (either inserted, missing, or replaced). These spans are then recursively compared to identify differences on word level. Using handcrafted rules, the differences are converted into edit operations that transform incorrect words appearing in transcribed texts into correct words present in the reference utterances. The set of edit operations is modeled after similar sets presented in previous works on ASR error correction (Ziętkiewicz, 2020, 2022; Kubis et al., 2022). Types of operations, together with examples, are shown in Table 4. The impact of the change in

the NLU outcome between the reference utterance and its back-transcribed counterpart is assessed in accordance with the criteria given in Section 3.2.

Having a method for identifying differences between reference utterances ($U$) and their back-transcriptions ($bt(U)$) and a set of guidelines for assessing the impact of the change in the NLU outcome as either positive or negative, we build a logistic regression model with the goal to predict if the extracted edit operations deteriorate the robustness of the model ($Y = 1$) or not ($Y = 0$) on the basis of the extracted edit operations ($editops$).

$$Y \sim editops(U, bt(U))$$

Afterward, we assess the impact of speech recognition errors on the robustness of the NLU model by extracting the regression coefficients that correspond to the edit operations that transform correct utterances into incorrect ones.

Framing the problem as a supervised classification task has several advantages. First, it allows us to incorporate any combination of the criteria outlined in Section 3.2 into the detection process.

| NLU model | TTS model | Metric | before BT | after BT | Δ |
|-----------|-----------|--------|-----------|----------|------|
| domain | FastSpeech | accuracy | 0.92 | 0.88 | -0.04 |
| intent | FastSpeech | accuracy | 0.88 | 0.82 | -0.05 |
| slots | FastSpeech | micro F1 | 0.80 | 0.61 | -0.19 |
| domain | Tacotron | accuracy | 0.92 | 0.89 | -0.03 |
| intent | Tacotron | accuracy | 0.88 | 0.84 | -0.04 |
| slots | Tacotron | micro F1 | 0.80 | 0.61 | -0.19 |

Table 5: NLU models performance.

Second, it allows us to consider different dimensions of the semantic representation of an NLU command, such as domain, intent, and slot values, either separately or in conjunction, enabling the evaluation of joint NLU models. Third, any classification method that quantifies the importance of the features specified at the input can be used to study the impact of speech recognition errors on the robustness of the NLU model. We rely on logistic regression because the regression coefficients are easy to interpret and the logistic model fits well to the provided data. However, a more elaborate model such as gradient boosted trees (Friedman, 2001) could be used instead to prioritize speech recognition errors by feature impurity.

## 4 Experiments

### 4.1 Data

Given that the back transcription technique does not require spoken data on the input, we decided to use the MASSIVE dataset (FitzGerald et al., 2022) to conduct the experiments. MASSIVE is a multilingual dataset for evaluating virtual assistants created on the basis of SLURP (Bastianelli et al., 2020). It consists of 18 domains, 60 intents, and 55 slots. The English subset of MASSIVE includes 11515 train utterances, 2033 development utterances, and 2974 test utterances[2]. MASSIVE was designed in such a way that test utterances are well separated semantically and syntactically from the trainset, and therefore, the results of state-of-the-art models do not exceed 90% F1-score with just hyper-parameters fine-tuning. This presents a good opportunity to test augmentation techniques like the one presented in this paper. We split data into train, validation, and test sets following the partition provided by MASSIVE. For learning NLU models, we use the train split. Finally, the test set is used to evaluate NLU models with respect to initial

and augmented data and to measure the impact of back transcription on the NLU performance.

### 4.2 Models

#### 4.2.1 Natural Language Understanding

We trained three separate XLM-RoBERTa (Conneau et al., 2020) models for three separate tasks: domain classification (Kubis et al., 2023a), intent classification (Kubis et al., 2023b) and slot filling (Kubis et al., 2023c). Although joint models present advantages over independent models (Zhang et al., 2019), since these three tasks are interdependent, we chose to train separate models to explore the impact of adversarial examples on each of these models separately. Following FitzGerald et al. (2022), we adopted the XLM-RoBERTa model architecture and fine-tuned the models on the MASSIVE dataset. We chose this model because it can be compared to models presented in MASSIVE and achieves better results in a multilingual setting when compared to mBERT (multilingual BERT). We also considered DeBERTaV3 (He et al., 2021), which outperforms both mBERT and XLM-RoBERTa on most NLP tasks, but when tested on the MASSIVE dataset, it did perform worse.

As a base model for all tasks, we used the pre-trained version of multilingual XLM-RoBERTa that was trained on 2.5TB of filtered Common-Crawl data containing 100 languages. Adam (Kingma and Ba, 2015) was used for optimization with an initial learning rate of $2e - 5$. The Domain and Intent models were trained for 5 epochs and the Slot model was trained for 20 epochs. For the Intent and Domain models, we tested scenarios where models were fine-tuned for more than 5 epochs, but none of them brought big improvements, while at the same time, this could decrease model generalization powers. In contrast, FitzGerald et al. (2022) experimented with model training between 3 and 30 epochs, which means that our models are tuned light-weighed, but as seen in the zero-

---

[2]Dataset version 1.1 from https://huggingface.co/datasets/AmazonScience/massive

shot scenario presented in the MASSIVE paper, the XLM-RoBERTa has remarkably good baseline generalization power, therefore we tried not to overfit models to data which would harm measuring the robustness of speech recognition models.

Table 5 reports the performance of NLU models measured with standard evaluation metrics before and after back transcription (*BT*) is applied to the MASSIVE dataset. One can observe that the NLU models that we trained for the experiments demonstrate state-of-the-art performance with respect to the standard evaluation measures before the back transcription procedure is applied.

### 4.2.2 Speech Processing

Our evaluation method relies on a combination of speech synthesis and automatic speech recognition models. For speech synthesis, we use two models. The first, FastSpeech 2 (Ren et al., 2021), is a non-autoregressive neural TTS model trained with the Fairseq $S^2$ toolkit (Wang et al., 2021) on the LJ Speech dataset[3] (part of LibriVox, Ito and Johnson, 2017). Text input is converted to phoneme sequences using the g2pE (Park and Kim, 2019) grapheme-to-phoneme library. Phoneme sequences are fed to the Transformer encoder, resulting in a hidden sequence, which is then enriched with a variance adaptor using duration, pitch, and energy information. The hidden sequence is then decoded with a Transformer decoder into mel spectrograms and waveforms with HiFi-GAN vocoder (Kong et al., 2020). The second TTS model is Tacotron 2[4] (Shen et al., 2018), trained on the same dataset (LJ Speech) and using the same vocoder (HiFi-GAN). In contrast to FastSpeech 2, Tacotron 2 is an autoregressive model trained directly on text input. Prediction of mel spectrograms is performed in a sequence-to-sequence manner using recurrent neural networks with attention. Automatic speech recognition is performed with Whisper[5] (Radford et al., 2022), a weakly supervised model trained on a massive collection of 680,000 hours of labeled audio data from diverse sources. The model is reported to generalize well on out-of-distribution datasets in training data and to be robust to domain changes and noise addition, making it a challenging choice for our method.

---

[3] https://huggingface.co/facebook/fastspeech2-en-ljspeech
[4] https://huggingface.co/speechbrain/tts-tacotron2-ljspeech
[5] https://huggingface.co/openai/whisper-large

### 4.3 Error Analysis

We report the robustness scores determined using the metrics proposed in Section 3.2 for the NLU models under study in Table 3. The results show that the evaluated models are far from perfect, even if we consider the most permissive metrics such as $R_1$, which neglects $I \rightarrow I$ and $I \rightarrow C$ changes, and $R_{13+}$, which rewards $I \rightarrow C$. The poor performance of slot models suggests that the accuracy-based measures of robustness that qualify the whole sample as incorrect if any slot value changes to an incorrect one due to back transcription may be too restrictive. Devising more fine-grained robustness measures for slot models is one of the issues that we plan to investigate in the future. Table 6 gives insight into the impact of the formulated robustness criteria on the discussed measures. Considering that $I \rightarrow I$ samples are responsible for 9%–30% of changes in the NLU outcome after back transcription and that $I \rightarrow C$ samples cause up to 10% of changes, it is clear that these criteria should not be neglected without a deliberate decision.

A qualitative comparison of the top 20 most frequent speech recognition errors demonstrated in Table 7 with the top 20 errors determined by the error detection model that treats all three criteria as negative (counterpart of metric $R_{123}$) presented in Table 8 shows that the rankings differ significantly. There are between 14 and 16 errors that appear in the top 20 lists of the error detection model and are not present on the corresponding lists of the most frequent errors, e.g.: *max[replace_macs]*, which is the result of a homophone substitution; and *""[add_before_dot]*, which indicates the problems caused by the missing period at the end of the utterance.

### 4.4 Quality of Synthesized Audio

We also checked if the overall quality of the synthesized audio is acceptable. For this purpose, we back-transcribed the dataset with both TTS models. Then, for each TTS model, we randomly sampled 10% of the prompts for which the result of back transcription differed from the input. The back-transcribed prompts were presented to the annotator along with the original prompts and the recording of the TTS output. The goal of the annotator was to choose which of the two transcripts was closer to the content of the recording. The order of the options was randomized so that the annotator did not know which was the original prompt and

| NLU model | TTS model | $C \to I$ | $I \to I$ | $I \to C$ | $Const$ |
|-----------|-----------|-----------|-----------|-----------|---------|
| domain | FastSpeech | 133 | 14 | 16 | 2811 |
| intent | FastSpeech | 176 | 36 | 16 | 2746 |
| slots | FastSpeech | 507 | 227 | 26 | 2214 |
| domain | Tacotron | 104 | 19 | 10 | 2841 |
| intent | Tacotron | 134 | 37 | 17 | 2786 |
| slots | Tacotron | 509 | 233 | 26 | 2206 |

Table 6: The number of NLU outcomes that changed due to back transcription or remained constant ($Const$).

| FastSpeech | Tacotron |
|------------|----------|
| ollie[replace_olly] | mail[add_prefix_e] |
| mail[add_prefix_e] | a[del] |
| pm[replace_m.] | ollie[replace_olly] |
| at[replace_add] | pm[replace_m.] |
| any[del_suffix_1] | only[sreplace_n_l] |
| 10[replace_ten] | in[del] |
| and[replace_in] | mails[add_prefix_e] |
| to[del] | i[del] |
| a[del] | at[replace_add] |
| only[sreplace_n_l] | a[add_suffix_n] |
| 9am[del_prefix_1] | 4[replace_four] |
| cnn[replace_n.] | 10[replace_ten] |
| he[add_suffix_y] | at[add_after_six] |
| at[add_after_six] | and[del] |
| and[del] | all[replace_olly] |
| 4[replace_four] | oli[replace_olly] |
| 6am[del_prefix_1] | to-do[split_on_first_-] |
| i'll[replace_olly] | light[add_suffix_s] |
| his[del_prefix_1] | the[del] |
| today's[sreplace_y'_y] | today's[sreplace_y'_y] |

Table 7: Top 20 most frequent errors.

| FastSpeech | Tacotron |
|------------|----------|
| pm[replace_m.] | pm[replace_m.] |
| chants[replace_suffix_ce] | ""[add_before_dot] |
| bowl[sreplace_w_i] | emmy[replace_amy] |
| cnn[replace_n.] | to-do[split_on_first_-] |
| inner[replace_inr] | 1[replace_one] |
| 9am[del_prefix_1] | paul's[sreplace_u_we] |
| dollar[add_before_us] | rare[replace_ray] |
| jeff[add_suffix_rey] | may[replace_email] |
| reburnet[replace_burnette] | today's[sreplace_y'_y] |
| at[add_after_six] | a[join_] |
| lets[replace_let's] | lice[del] |
| 6am[del_prefix_1] | at[add_after_six] |
| today's[sreplace_y'_y] | and[replace_suffix_y] |
| max[replace_macs] | enlightening[replace_lighting] |
| natie[replace_naty] | pondicherry[sreplace_rr_r] |
| sassy[replace_prefix_c] | barn[del_suffix_1] |
| pondicherry[sreplace_rr_r] | yuli[add_suffix_a] |
| ordered[del_suffix_2] | by[del] |
| mr[add_suffix_.] | will[del] |
| it[replace_a] | 430[replace_thirty] |

Table 8: Top 20 errors that deteriorate $R_{123}$.

which was the back-transcribed one. If both options were equally viable, the annotator was allowed to choose *both* as the answer.

Table 9 presents sample sizes (*total*), the number of TTS outputs recognized as closer to the original prompt (*utt*), closer to the back-transcribed text (*aug*) or marked as close to *both*. The last column shows the percentage of TTS outputs that resemble the original prompt (i.e., $\frac{utt+both}{total}$). The results show that about 85% of TTS-generated readings of the selected prompts were good enough to at least such an extent that the annotator indicated them to be either equally close or closer to the original prompt than the recognized text, which confirms the acceptable quality of the synthesis. Furthermore, both TTS models have reached similar results, which implies that they can be used interchangeably.

## 4.5 Robustness Scores for Voice Recordings

To confirm that TTS-generated speech samples can be used in place of voice recordings, we verified that the robustness scores obtained for the synthesized samples are similar to the scores obtained for the recordings. For this purpose, we conducted an experiment using audio samples from the SLURP dataset (Bastianelli et al., 2020), which contains recordings of over a hundred speakers gathered in acoustic conditions that match a typical home/office environment with varying locations and directions of speakers toward the microphone array. Thus, the recordings we use for evaluation are not overwhelmingly noisy, but they provide a realistic use case for a virtual assistant.

First, we applied the back transcription procedure to the text prompts extracted from SLURP. Next, we ran the ASR model on the audio recordings corresponding to the extracted prompts and applied the NLU models to the transcribed texts. Finally, we compared the robustness scores calculated for back-transcribed and transcribed texts. As shown in Table 10, the robustness scores obtained on TTS-generated speech samples closely resemble those obtained on real speech samples from SLURP. The difference between scores is 0.03 on average and not greater than 0.08, which confirms

| TTS | total | utt | aug | both | resemblance |
|---|---|---|---|---|---|
| Tacotron | 121 | 73 | 19 | 29 | 84.30% |
| FastSpeech | 115 | 66 | 16 | 33 | 86.09% |

Table 9: TTS evaluation results.

| NLU model | Audio | $R_{123}$ | $R_{13}$ | $R_{12}$ | $R_1$ | $R_{123+}$ | $R_{13+}$ |
|---|---|---|---|---|---|---|---|
| domain | SLURP | 0.8675 | 0.8755 | 0.8786 | 0.8875 | 0.8802 | 0.8890 |
| intent | SLURP | 0.8209 | 0.8402 | 0.8348 | 0.8562 | 0.8375 | 0.8588 |
| slots | SLURP | 0.4308 | 0.4813 | 0.4413 | 0.4968 | 0.4547 | 0.5126 |
| domain | FastSpeech | 0.8590 | 0.8642 | 0.8710 | 0.8770 | 0.8727 | 0.8788 |
| intent | FastSpeech | 0.7962 | 0.8084 | 0.8137 | 0.8283 | 0.8177 | 0.8324 |
| slots | FastSpeech | 0.3629 | 0.4162 | 0.3705 | 0.4281 | 0.3835 | 0.4439 |
| domain | Tacotron | 0.8879 | 0.8966 | 0.8966 | 0.9059 | 0.8976 | 0.9069 |
| intent | Tacotron | 0.8302 | 0.8475 | 0.8446 | 0.8638 | 0.8473 | 0.8664 |
| slots | Tacotron | 0.3875 | 0.4476 | 0.3945 | 0.4585 | 0.4054 | 0.4714 |

Table 10: NLU models robustness determined with SLURP.

that using TTS-generated speech samples in place of audio recordings is justified, taking into consideration that Whisper ASR achieves the word error rates of 0.1625, 0.1121 and 0.1165 for SLURP, Tacotron and Fastspeech samples, respectively.

## 5 Conclusion

In this paper, we proposed a method for assessing the robustness of NLU models to speech recognition errors. The method repurposes the NLU data used for model training and does not depend on the availability of spoken corpora. We introduced criteria for robustness that rely on the outcome of the NLU model but do not assume any particular semantic representation of the utterances. We showed how these criteria can be used to formulate summary metrics and constructed an analytical model that prioritizes individual categories of speech recognition errors on the basis of their impact on the (non-)robustness of the NLU model. Finally, we performed an experimental evaluation of the robustness of Transformer-based models and investigated the impact of using text-to-speech models in place of audio recording.

## Limitations

The presented method compares input utterances with the same input synthesized by TTS and processed by ASR. This setting introduces two limitations for the NLU component. First, the architecture, the training data, and finally, the quality of TTS and ASR systems impact generated data variation. Configuration of different systems might generate different outcomes, so it is more difficult to draw general conclusions for other TTS and ASR systems used. Additionally, the set of possible variations will be, in most cases, limited to errors produced by TTS and ASR. While those types of errors were the primary goal of this article, it could be extended to measure further NLU robustness problems.

## Acknowledgements

This research was partially funded by the *CAIMAC: Conversational AI Multilingual Augmentation and Compression* project, a cooperation between Adam Mickiewicz University and Samsung Electronics Poland.

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

## A Wav2Vec Experiments

Due to the limited space the main text presents only the back transcription experiments conducted with the use of Whisper (Radford et al., 2022) ASR model. However, in order to verify that the presented evaluation method generalizes to other speech recognition models we also carried out experiments with Wav2vec 2.0[6] (Baevski et al., 2020), a model pre-trained in a self-supervised manner on audio-only data from LibriVox (LV60k) using masking of latent speech representation and fine-tuned on labeled data (LibriSpeech 960h, Panayotov et al., 2015) for the speech recognition task. Following the main text we report the robustness results determined with the use of Wav2Vec model in Table 11 and 12 and present back transcription evaluation results in Table 13 and 14.

---

[6] https://huggingface.co/facebook/wav2vec2-base-960h

| NLU model | TTS model | $R_{123}$ | $R_{13}$ | $R_{12}$ | $R_1$ | $R_{123+}$ | $R_{13+}$ |
|---|---|---|---|---|---|---|---|
| domain | FastSpeech | 0.7829 | 0.7884 | 0.7906 | 0.7967 | 0.7926 | 0.7988 |
| intent | FastSpeech | 0.6903 | 0.7030 | 0.7016 | 0.7159 | 0.7065 | 0.7210 |
| slots | FastSpeech | 0.3037 | 0.3436 | 0.3085 | 0.3506 | 0.3194 | 0.3634 |
| domain | Tacotron | 0.8050 | 0.8141 | 0.8156 | 0.8256 | 0.8181 | 0.8281 |
| intent | Tacotron | 0.7254 | 0.7379 | 0.7389 | 0.7533 | 0.7437 | 0.7583 |
| slots | Tacotron | 0.3050 | 0.3473 | 0.3103 | 0.3549 | 0.3219 | 0.3688 |

Table 11: NLU models robustness.

| NLU model | TTS model | $C \rightarrow I$ | $I \rightarrow I$ | $I \rightarrow C$ | $Const$ |
|---|---|---|---|---|---|
| domain | FastSpeech | 407 | 43 | 21 | 2503 |
| intent | FastSpeech | 543 | 94 | 35 | 2302 |
| slots | FastSpeech | 1093 | 384 | 34 | 1463 |
| domain | Tacotron | 332 | 45 | 27 | 2570 |
| intent | Tacotron | 449 | 82 | 38 | 2405 |
| slots | Tacotron | 1027 | 378 | 35 | 1534 |

Table 12: The number of NLU outcomes that changed due to back transcription or remained constant ($Const$).

| TTS | total | utt | aug | both | utt+both | percentage |
|---|---|---|---|---|---|---|
| Tacotron | 207 | 171 | 19 | 17 | 188 | 90.82% |
| FastSpeech | 217 | 183 | 18 | 16 | 199 | 91.71% |

Table 13: TTS evaluation results.

| NLU model | Audio | $R_{123}$ | $R_{13}$ | $R_{12}$ | $R_1$ | $R_{123+}$ | $R_{13+}$ |
|---|---|---|---|---|---|---|---|
| domain | SLURP | 0.6539 | 0.6590 | 0.6642 | 0.6700 | 0.6694 | 0.6755 |
| intent | SLURP | 0.5910 | 0.6012 | 0.6014 | 0.6131 | 0.6083 | 0.6206 |
| slots | SLURP | 0.2943 | 0.3304 | 0.3004 | 0.3391 | 0.3144 | 0.3561 |
| domain | FastSpeech | 0.7844 | 0.7901 | 0.7924 | 0.7988 | 0.7945 | 0.8010 |
| intent | FastSpeech | 0.7007 | 0.7089 | 0.7155 | 0.7256 | 0.7214 | 0.7319 |
| slots | FastSpeech | 0.3016 | 0.3428 | 0.3061 | 0.3495 | 0.3163 | 0.3618 |
| domain | Tacotron | 0.8024 | 0.8085 | 0.8130 | 0.8200 | 0.8155 | 0.8225 |
| intent | Tacotron | 0.7251 | 0.7373 | 0.7405 | 0.7548 | 0.7459 | 0.7605 |
| slots | Tacotron | 0.3034 | 0.3425 | 0.3086 | 0.3502 | 0.3203 | 0.3644 |

Table 14: NLU models robustness determined with SLURP.