# OpenReview forum: "Back Transcription as a Method for Evaluating Robustness of Natural Language Understanding Models to Speech Recognition Errors"
_EMNLP/2023/Conference — EMNLP 2023 Main_

### Official Review · Reviewer_ki6r · 2023-07-27

**Soundness:** 2

**Excitement:**

2: Mediocre: This paper makes marginal contributions (vs non-contemporaneous work), so I would rather not see it in the conference.

**Paper Topic And Main Contributions:**

This paper uses a speech synthesiser to assess the impact of asr errors on a semantic decoder designed to extract domain/intent/slot values from natural language utterances.  Given a set of NLU training texts, each text is synthesised and then recognised to give a second possibly corrupted version of the text.  Various error metrics are proposed depending on whether the asr error generates an actual semantic error or serendipitously corrects an error in the source text.   In addition, heuristics are used to find a set of edit operations which restore the asr error.  Logistic regression is then used to relate the edit operation to its impact on NLU robustness.

Experiments are conducted using the MASSIVE NLU dataset with XLM-RoBERTa decoders.  Two synthesisers, FastSpeech and Tacotron are tested alongside Whisper ASR.  Also naturally spoken utterances are extracted from SLURP to confirm that TTS input is representative of human speech input.  The results show that where errors occur in the source text, the ASR actually corrects a substantial number of them so simple NLU error rates do not tell the whole story.  They also claim that edit operation regression allows them to identify the type of ASR error that should be prioritised.


**Questions For The Authors:**

It would be good to provide ASR error rates.

If you could identify a specific type of ASR error which is impacting on NLU performance, how would you fix it?

Some ASR developers are now using TTS to augment their training data.  How would this impact on your analysis method?


**Reasons To Accept:**

The paper is clearly written and presented and it provides a practical method of leveraging an NLU training set to also determine robustness to ASR errors.  The paper also provides some evidence that modern TTS systems such as Tacotron provide reasonable surrogates for natural speech in testing an ASR system.


**Reasons To Reject:**

There is nothing very original in the paper.   The authors do provide a way of automatically generating analytics for an ASR-NLU tandem but they do not demonstrate any particular insights that are gained from the study.  Table 8 provides a large set of numbers to 4 figure accuracy, but its not clear to me that there is any information which could not have been deduced from simple ASR WER rates and standard NLU F1 measures.  The paper would be much more compelling if WER/F1 rates had been provided, and the additional value of the proposed metrics clearly demonstrated.


**Reproducibility:**

4: Could mostly reproduce the results, but there may be some variation because of sample variance or minor variations in their interpretation of the protocol or method.

**Reviewer Confidence:**

4: Quite sure. I tried to check the important points carefully. It's unlikely, though conceivable, that I missed something that should affect my ratings.

**Typos Grammar Style And Presentation Improvements:**

Line 461 qualify -> quality ?

---

> ### Author Rebuttal · Authors · 2023-08-25
>
> It is stated in the review that "Table 8 provides a large set of numbers to 4 figure accuracy, but its not clear to me that there is any information which could not have been deduced from simple ASR WER rates and standard NLU F1 measures. The paper would be much more compelling if WER/F1 rates had been provided, and the additional value of the proposed metrics clearly demonstrated."
>
> In order to clarify on this issue let us analyze three cases:
>
> 1. Can ASR WER be used in isolation to deduce the robustness of an NLU model?
>
>     ASR WER does not track the NLU outcome. Hence, it cannot be used to deduce NLU performance.
>
> 2. Can ASR WER be used in combination with F1 measure to deduce the robustness of an NLU model?
>
>     No. Unfortunately, due to the limited size of the paper we postponed the analysis of F1 measure to Appendix B. Table 13 shows how the building blocks of F-measure change due to the changes in NLU outcome between the reference utterances and their back-transcribed counterparts. One can observe that for each category of changes that we distinguished in Section 3.1 some components of F-measure increase (↑) while others decrease (↓) or remain unchanged (=). Therefore, measuring the difference in F-scores between reference utterances and back-transcribed texts leads to results that cannot be interpreted in a meaningful way. This is also the reason why we refrained from reporting delta F1 scores in the main part of the paper. However, F-scores for NLU models are shows in Table 14 (Appendix C). We will move the analysis of F-score from Appendix B to the main part to make it clear in the final version of the paper.
>
> 3. Can ASR WER be used in combination with Exact Match Accuracy to deduce the robustness of an NLU model?
>
>     Yes. However, it is just one of six alternative measures (i.e. R_13+) that can be used to study the robustness of an NLU model in terms of the criteria outlined in Section 3.1. In Section 3.2 (lines 222-265) we provide **the insights** on using these alternative measures. Just to summarize:
>
>     a) R13+ measure neglects I → I changes and treats I → C changes as positive. Thus this approach is sufficient for testing an NLU model in isolation, but it does not take into account that the behaviour of downstream modules of a dialogue system that consume the outcome of an NLU model can deteriorate due to the change in labelling of incorrect results.
>
>     b) R123+ promotes changing incorrect outcomes to correct ones which is reasonable, if we assume that the downstream module behaves correctly when presented with correct input.
>
>     c) If the downstream module relies on the outcome of NLU regardless of its status, then R123 should be preferred in place of R123+.
>
>     d) R12 which penalizes changes between incorrect labels, but neglects the impact of C → I changes is a rational choice for assessment of an NLU model that precedes a downstream module dedicated to correcting incorrect NLU outcomes such as a rule-based post-processor.
>
>     e) R1 which penalizes the drop in accuracy, but neglects any other changes is a metric that tracks the volume of samples that become incorrect due to the use of an ASR system. Therefore, it is suitable for monitoring the regressions of the ASR-NLU pair across consecutive revisions of the ASR model.
>
>     f) The penalization of positive changes by R13 makes this metric also a reasonable choice for tracking robustness of NLU models that should act consistently in presence of reference texts and their transcribed counterparts. This is the case of NLU models that are designed to handle both the input typed by the user and the input that comes from an ASR system.
>
>     g) The same holds for R123 which contrary to R13 also takes into account the impact of I → I changes on downstream modules of a dialogue system.
>
> As for the questions:
>
> Q1. It would be good to provide ASR error rates.
>
> The ASR WER results for the entire SLURP dataset are as follows:
>
> 1. Audio: 0.1625
> 2. Tacotron: 0.1121
> 3. Fastspeech: 0.1165
>
> The ASR WER results for utterances that change in NLU outcome due to back-transcription are as follows:
>
> 1. Audio: 0.4842
> 2. Tacotron: 0.4362
> 3. Fastspeech: 0.4758
>
> We will report WER in the final version of the paper.
>
> Q2. If you could identify a specific type of ASR error which is impacting on NLU performance, how would you fix it?
>
> For this purpose we can utilize the back-translated dataset to select samples to be included in the NLU model training/augmentation dataset. If you are interested in fixing a specific error, you can extract from the back-transcribed dataset hypotheses h(s) that contain this error along with the expected NLU outcomes e(s) and include them in the NLU training dataset.
>
> The aforementioned procedure combined with the method for detecting speech recognition errors presented in Section 3.3 can be used to form the NLU training/augmentation dataset that addresses the errors that deteriorate the robustness of the NLU model in the most significant way.
>
> Q3. Some ASR developers are now using TTS to augment their training data. How would this impact on your analysis method?
>
> The presented method does not rely on the use of a specific TTS model, hence one can counter this problem by selecting a TTS that was not used for augmenting ASR training data.

---

### Official Review · Reviewer_jqhQ · 2023-08-05

**Soundness:** 4

**Excitement:**

4: Strong: This paper deepens the understanding of some phenomenon or lowers the barriers to an existing research direction.

**Paper Topic And Main Contributions:**

This paper addresses the problem of assessing the robustness of Natural Language Understanding (NLU) models to speech recognition errors. The main contribution of the paper is the proposal of a method to evaluate NLU model robustness without the need for spoken data input. Instead, the method repurposes the NLU training data and uses Text-to-Speech (TTS) models to generate synthetic utterances. These synthetic utterances are then processed by Automatic Speech Recognition (ASR) models to simulate speech recognition errors. By comparing the NLU model's performance on the original text and the TTS-generated text, the paper quantifies the impact of speech recognition errors on the NLU outcomes.

The paper introduces criteria for robustness that rely on the outcome of the NLU model without assuming any specific semantic representation. It formulates summary metrics based on these criteria to prioritize different categories of speech recognition errors based on their impact on NLU model robustness. The paper also evaluates Transformer-based NLU models using this method and investigates the use of TTS-generated speech samples in place of audio recordings.

**Questions For The Authors:**

1. Have you considered evaluating the robustness of NLU models on real spoken data rather than relying solely on TTS-generated speech samples? How do you believe the findings might differ in such a scenario?

2. The paper mainly focuses on Transformer-based models like FastSpeech and Tacotron. Have you considered comparing the robustness of NLU models with other NLU architectures or models? How do you think different architectures might perform in terms of robustness?

3. The experiments are conducted on the MASSIVE dataset, which comprises 18 domains. Could you provide insights into the domain diversity and the potential implications of the findings on NLU robustness for other domains not covered by the dataset?

4. The paper introduces robustness criteria and summary metrics. Have you considered incorporating additional evaluation metrics to capture other aspects of NLU model performance and robustness?

5. While TTS-generated speech samples show reasonable resemblance to real speech, have you conducted any user studies or surveys to assess the perceptual quality and naturalness of the synthesized utterances? How confident are you that the TTS models sufficiently capture the nuances of real speech?

**Reasons To Accept:**

1. Novel Methodology: The paper presents a novel and creative method for evaluating NLU model robustness in the presence of speech recognition errors. By repurposing NLU training data and using TTS models to generate synthetic utterances, the paper overcomes the need for spoken data input, making it easier for researchers to study NLU robustness.

2. Broad Applicability: The proposed method is not tied to any specific semantic representation, allowing it to be applied to various NLU models and architectures. This broad applicability makes it relevant to a wide range of NLP tasks and scenarios.

3. Data Efficiency: The method utilizes existing NLU training data, which saves computational resources and reduces the need for additional labeled spoken data, especially for low-resource languages or domains where obtaining spoken data can be challenging.

4. Insights into NLU Robustness: By quantifying the impact of speech recognition errors on NLU outcomes, the paper provides valuable insights into the robustness of Transformer-based NLU models. These insights can help researchers and practitioners understand how NLU models perform in real-world scenarios with imperfect speech recognition input.

5. Potential for Further Research: The paper opens up avenues for further research in NLU model robustness assessment. The proposed criteria and summary metrics can be extended and adapted to investigate other sources of errors and vulnerabilities in NLU models.

**Reasons To Reject:**

1. Limited Comparison: The paper mainly focuses on comparing Transformer-based models, specifically FastSpeech and Tacotron, for NLU model robustness. While this is insightful, a broader comparison with other NLU architectures or models would strengthen the paper's impact and generalizability.

2. Overemphasis on TTS Models: The paper heavily relies on TTS models for generating synthetic utterances to evaluate NLU model robustness. While TTS-generated speech samples show reasonable resemblance to real speech, there may be inherent limitations in TTS systems, such as prosody and naturalness, which could affect the robustness assessment.

3. Lack of Evaluation on Real Spoken Data: The paper primarily evaluates the NLU models on synthetic utterances generated by TTS models. While this approach is efficient, the absence of evaluation on real spoken data might limit the real-world applicability and generalizability of the findings.

4. Limited Discussion on Error Types: The paper discusses specific types of speech recognition errors, such as insertions, deletions, and replacements, but it lacks a comprehensive analysis of all possible error types, which could provide a more holistic understanding of NLU robustness.

5. Limited Domain Diversity: The experiments conducted on the MASSIVE dataset might lack domain diversity, potentially limiting the generalizability of the findings to other domains and real-world applications.

**Reproducibility:**

3: Could reproduce the results with some difficulty. The settings of parameters are underspecified or subjectively determined; the training/evaluation data are not widely available.

**Reviewer Confidence:**

3: Pretty sure, but there's a chance I missed something. Although I have a good feel for this area in general, I did not carefully check the paper's details, e.g., the math, experimental design, or novelty.

---

> ### Author Rebuttal · Authors · 2023-08-26
>
> Please find below the answers to your questions:
>
> Q1: Have you considered evaluating the robustness of NLU models on real spoken data rather than relying solely on TTS-generated speech samples? How do you believe the findings might differ in such a scenario?
>
> Yes, we performed evaluation with the use of real spoken data from the SLURP dataset. We show the results in Table 8 (Section 4.5).
>
> Q2: The paper mainly focuses on Transformer-based models like FastSpeech and Tacotron. Have you considered comparing the robustness of NLU models with other NLU architectures or models? How do you think different architectures might perform in terms of robustness?
>
> Taking into consideration that Transformer-based models exhibit state-of-the-art performance in the NLU task, we think that the other NLU architectures will perform worse in the terms of robustness.
> Unfortunately, due to time constraints and limited space we focused solely on Transformer-based models, which are currently the most popular SOTA method.
>
> Q3: The experiments are conducted on the MASSIVE dataset, which comprises 18 domains. Could you provide insights into the domain diversity and the potential implications of the findings on NLU robustness for other domains not covered by the dataset?
>
> MASSIVE and SLURP cover utterances spanning 18 domains, 60 intents, and 55 slots. The range of covered domains is broad, it includes calendar, weather, transport, cooking, etc., as well as general topics.
> Taking into consideration that several categories of ASR errors occur across different domains in our data, it is reasonable to assume that the robustness for other domains will be at least partially related to the robustness determined for the domains under study.
>
> Q4: The paper introduces robustness criteria and summary metrics. Have you considered incorporating additional evaluation metrics to capture other aspects of NLU model performance and robustness?
>
> The method for detecting speech recognition errors outlined in Section 3.3 can be adapted to other categories of errors. We consider generalizing it to errors that are not easily summarized by edit operations (e.g. restarts in spontaneous speech).
>
> Q5: While TTS-generated speech samples show reasonable resemblance to real speech, have you conducted any user studies or surveys to assess the perceptual quality and naturalness of the synthesized utterances? How confident are you that the TTS models sufficiently capture the nuances of real speech?
>
> We present a study on quality of synthesized audio in Section 4.4.

---

### Official Review · Reviewer_PdVj · 2023-08-11

**Soundness:** 4

**Excitement:**

4: Strong: This paper deepens the understanding of some phenomenon or lowers the barriers to an existing research direction.

**Paper Topic And Main Contributions:**

The paper proposes a new method to investigate the robustness of natural language understanding (NLU) models against speech recognition errors. The problem they tackle is the lack of assessment procedure for NLU systems. The proposed method that is based on the back transcription can be split into three steps: back transcription, robustness assessment and error analysis. The back transcription module takes a reference text, generates audio and its transcription. The reference and the transcription are the output of this module. The robustness assessment evaluates the model prediction using the expected result from reference text and the predicted result from transcription. For the assessment, they investigated three classes of changes. The choice of the criterias depends on how to penalize the model. Finally, the error analysis identifies the differences between the reference and transcription by aligning them. They build a logistic regression model to predict the edit operations between the two sequences. The experimental results showed that using generated audio samples vs recordings didn’t represent much differences.

**Questions For The Authors:**

a.For the speech processing part, the authors use the pre-trained models for both TTS and ASR tasks. Were these models trained on any data from the dataset? What is the performance of these models? How noisy are the generated transcriptions?  For TTS generated audios, is there only human evaluation explained in Section 4.4?

b. Can changing an ASR system resulting noisier transcript be more dramatic to observe the model robustness?

**Reasons To Accept:**

The paper investigates the robustness of NLU using a back transcription method and showed that using synthetic audio didn’t change the result compared to using recording. This findings is a proof to show that labeled audio data is not necessary to investigate the robustness. Using several TTS models also supports the claim, since they don’t possess much differences in the result. Although the authors use the XLM-Roberta model to test their proposition, the method is modular so that each method used in the modules can be changed.

**Reasons To Reject:**

a. As indicated in Section 3, the proposed method requires a dataset with user utterances and semantic representation of the uttered commands. This requirement can limit the usage of the model especially with the low-resource languages.

b. The choice of XLM-Roberta model is not convincing, since the paper investigates the robustness of the NLU model, not the best performing one. Along with XML-Roberta, other models can be also investigated on the MASSIVE dataset. Also, seeing the results other than MASSIVE dataset could be more supportive.

**Reproducibility:**

4: Could mostly reproduce the results, but there may be some variation because of sample variance or minor variations in their interpretation of the protocol or method.

**Reviewer Confidence:**

4: Quite sure. I tried to check the important points carefully. It's unlikely, though conceivable, that I missed something that should affect my ratings.

---

> ### Author Rebuttal · Authors · 2023-08-26
>
> Please find below the answers to your questions:
>
> Q1. For the speech processing part, the authors use the pre-trained models for both TTS and ASR tasks. Were these models trained on any data from the dataset?
>
> The authors of TTS and ASR models that we use did not indicate MASSIVE or SLURP as a source of training data.
>
> Q2. What is the performance of these models?
>
> We selected state-of-the-art models for the tasks. The Whisper ASR model that we use achieved WER of 2.7 on LibriSpeech (https://cdn.openai.com/papers/whisper.pdf). The FastSpeech2 model achieved MOS of 4.18±0.06 on LJSpeech corpus (https://arxiv.org/pdf/2109.06912.pdf). Tacotron2 based models achieve MOS of up to 4.53 (https://arxiv.org/pdf/1712.05884.pdf).
>
> Q3. How noisy are the generated transcriptions?
>
> We did not inject additional noise into the synthesized samples. Hence, they have to be considered clean.
>
> Q4. For TTS generated audios, is there only human evaluation explained in Section 4.4?
>
> Yes, we considered the evaluation procedure presented in Section 4.4 to be sufficient taking into account that we use state-of-the-art pre-trained TTS models for the task.
>
> Q5. Can changing an ASR system resulting noisier transcript be more dramatic to observe the model robustness?
>
> Yes, indeed. Please check Appendix A were we report results for a weaker ASR system.

---

### Official Review · Reviewer_mLMm · 2023-08-12

**Soundness:** 2

**Excitement:**

2: Mediocre: This paper makes marginal contributions (vs non-contemporaneous work), so I would rather not see it in the conference.

**Paper Topic And Main Contributions:**

In this paper, the authors investigate the impact of speech recognition errors on natural language understanding system by using back transcription technique that transcribes TTS generated speech using an ASR. The technique is investigated to simulate speech recognition errors and evaluate the quality of predictions from NLU models.

**Questions For The Authors:**

What is the ASR WER performance for synthesized and original speech?

Will the authors make this dataset available to public?

**Reasons To Accept:**

The paper proposes a dataset creation technique called Back transcription generate additional data points for NLU tasks by transcribing synthesized speech. This technique can be replicated for several spoken language processing tasks to augment datasets for evaluation.

**Reasons To Reject:**

The main aim of the paper is unclear to me. It is common for speech recognition errors to persist and degrade the performance of NLU  impacting the intent, domain and slot predictions and therefore the overall conversational task. The authors seem to propose the technique to augment datasets for evaluation of NLU tasks. There is significant difference between synthesized speech and real speech and the quality of transcripts for each scenario. The quality of ASR transcripts of real speech is often impacted by background noise, L1 backgrounds, vocabulary usage, etc. while in TTS the speech can be from a fixed speaker. The authors do show however that there is very small difference in robustness between real and TTS generated speech. In my opinion, this should be surveyed in more detail and discussed in the paper.

In terms of the range of experiments and evaluations, I think there is limited work demonstrated in the paper and the novelty is minor. Using transformer based language models models for NLU modeling or Tacotron for speech synthesis is fairly common today. The only novelty is the dataset creation.

Further, no metrics for ASR quality (WER performance) for real and synthesized speech are shared in the paper.



**Reproducibility:**

3: Could reproduce the results with some difficulty. The settings of parameters are underspecified or subjectively determined; the training/evaluation data are not widely available.

**Reviewer Confidence:**

4: Quite sure. I tried to check the important points carefully. It's unlikely, though conceivable, that I missed something that should affect my ratings.

---

> ### Author Rebuttal · Authors · 2023-08-25
>
> As for "The authors do show however that there is very small difference in robustness between real and TTS generated speech. In my opinion, this should be surveyed in more detail and discussed in the paper." we agree with the reviewer that difference in robustness between real and TTS generated speech that we show is small. For the purpose of evaluating the proposed method (Sec. 4.5) we used the SLURP dataset which contains recordings of 100+ speakers gathered in acoustic conditions that match a typical home/office environment with varying locations and directions of speakers towards the mic array (cf. https://arxiv.org/pdf/2011.13205.pdf). Therefore, the recordings are not overwhelmingly noisy, but they provide a realistic use case for an intelligent personal assistant. Especially, taking into account that they encompass eighteen NLU domains. We will highlight the limitations of the dataset used for the study in the final version of the paper. We also welcome any suggestions for other open, multi-domain, spoken language understanding dataset that we can use along SLURP.
>
> As for "Using transformer based language models models for NLU modeling or Tacotron for speech synthesis is fairly common today. ",
> please notice that we present "a method for investigating the impact of speech recognition errors on the performance of natural language understanding models". Therefore, It would be a methodological error on our side, if we proposed both a new family of NLU or TTS models alongside a new method for evaluating NLU models. We deliberately used state-of-the-art open TTS, ASR and NLU models to have a reproducible environment for conducting the experiments. An environment which is unbiased by in-house research on NLU or TTS models.
>
> As for "The only novelty is the dataset creation.". Well, to be fair the dataset is not a novelty either. As in the case of TTS, ASR and NLU models we worked on well recognized datasets (MASSIVE/SLURP) with the intent of having a reproducible environment. However, the novel aspects of our method are as follows:
>
> 1. In section 3.2. w introduce criteria for robustness formed on the basis of reference and back-transcribed utterances.
> 2. In section 3.2 we construct a family of robustness metrics that exhibit different characteristics with regard to the modules that precede and follow the NLU model in a spoken dialogue system. We also show that a common practice of measuring the difference in accuracy for evaluating the impact of ASR on the NLU model is just one of six alternative options (R13+).
> 3. In section 3.3 we propose a method for detecting speech recognition errors on the basis of the presented robustness criteria.
>
> As for the questions: We will release the dataset on GitHub. Currently it is hidden due to anonymity requirements.
>
> The ASR WER results for the entire SLURP dataset are as follows:
> 1. Audio: 0.1625
> 2. Tacotron: 0.1121
> 3. Fastspeech: 0.1165
>
> The ASR WER results for utterances that change in NLU outcome due to back-transcription are as follows:
> 1. Audio: 0.4842
> 2. Tacotron: 0.4362
> 3. Fastspeech: 0.4758

---

### Meta-Review · Area_Chair_WXiU · 2023-09-18

**Recommendation:** 4

**Metareview:**

This work addresses the problem of assessing the robustness of Natural Language Understanding (NLU) models to speech recognition errors. The method proposed repurposes NLU training data and uses Text-to-Speech (TTS) models to generate synthetic utterances. These synthetic utterances are then processed by Automatic Speech Recognition (ASR) models to simulate speech recognition errors. By comparing the NLU model's performance on the original text and the TTS-generated text, the paper quantifies the impact of speech recognition errors on NLU.

Beyond automating combined ASR-NLU testing, one of the key contributions that the paper provides is in proposing a number of additional metrics that can be derived by this method and explaining the insights that they provide versus existing metrics. That said, more than one reviewer would prefer that WER/F1 rate be provided in the main text to make the argument for these new metrics more compelling.

---

### Decision · Program_Chairs · 2023-10-07

**Decision:**

Accept-Main

**Comment:**

This work addresses the problem of assessing the robustness of Natural Language Understanding (NLU) models to speech recognition errors. The method proposed repurposes NLU training data and uses Text-to-Speech (TTS) models to generate synthetic utterances. These synthetic utterances are then processed by Automatic Speech Recognition (ASR) models to simulate speech recognition errors. By comparing the NLU model's performance on the original text and the TTS-generated text, the paper quantifies the impact of speech recognition errors on NLU.

Beyond automating combined ASR-NLU testing, one of the key contributions that the paper provides is in proposing a number of additional metrics that can be derived by this method and explaining the insights that they provide versus existing metrics. That said, more than one reviewer would prefer that WER/F1 rate be provided in the main text to make the argument for these new metrics more compelling.